# Shared bikes distribution vehicle routing problem with split delivery considering carbon emission

Guanghui Chen[1]*, Huicong Li[2], Bing Su[1], Qinge Guo[1], Hao Ji[1]

1 School of Economics and Management, Xi'an Technological University, Xi'an, China, 2 Henan Light Industry Vocational College, Zhengzhou, China

* ghui_chen@163.com

## Abstract

The distribution of shared bikes is different from that of other goods. There are some demand stations which need a large number of shared bikes, such as bus stations, subway exits and business districts. The demand of these stations cannot be met in a single delivery, so the demand can be split into batches for distribution. Therefore, shared bikes need to be delivered from distribution centers to demand stations. However, these delivery vehicles generate carbon emissions during the process, which has an impact on environment. Thus, shared bikes distribution vehicle route selection with considering carbon emission under demand splitting is an important problem. The paper established a model for distribution vehicle route selection of shared bikes considering carbon emission which aims at minimization the sum of carbon emission cost and delivery cost, under demand splitting of the stations and delivery vehicles with load limit. Then an approximation algorithm *GA* is designed to solve it. The time complexity of *GA* was proved, and the upper and lower bounds of the approximate ratio of *GA* are discussed. Finally, an empirical example was facilitated by examining real shared bikes stations in the Yanta district of Xi'an, China, to verify the effectiveness of the model and algorithm. The approximation ratio of *GA* is 3.52 which shows that the approximate performance of the algorithm in the example is good. The results and conclusions yield a theoretical basis for decision-makers to optimize the delivery of shared bikes.

## 1. Introduction

The delivery of shared bikes differs from the delivery of other goods. At some demand stations that need a large number of shared bikes, it is impossible for a single delivery to meet all their needs. In such demand stations, the demand can be split into batches for distribution [1]. Shared bikes play a significant role in reducing urban congestion and lowering carbon emissions from transportation, but the delivery vehicles generate carbon emissions. Therefore, shared bikes distribution vehicle route

**Data availability statement:** The data supporting the findings of this study are available within the article.

**Funding:** This work was supported in part by the National Nature Science Foundation of China under Grant 52302443, and in part by the Social Science Foundation of Shaanxi Province under Grant 2023R059. The funders had no role in study design, data collection and analysis, decision to publish, or preparation of the manuscript.

**Competing interests:** The authors have declared that no competing interests exist.

selection with considering carbon emission under demand splitting is an important issue of great theoretical value and realistic meaning.

In practice, the public's demand for shared bikes varies across different stations. For instance, there is a huge demand at bus stops, subway entrances, and near commercial areas. To meet the public's normal travel needs, it is necessary to promptly replenish the shortage of shared bikes at these demand stations. However, a single delivery by a delivery vehicle cannot satisfy all the needs at such demand stations. Therefore, shared bikes are often delivered in split. If not split deliveries, some demand stations with significant demand close to the distribution center can receive timely bike replenish, but a single delivery cannot meet all their needs. Additionally, some demand stations distant to the distribution center with general demand are hard to cover. Thus leads to a small coverage area for public use of shared bikes. Moreover, the delivery vehicles generate carbon emission during distribution, which depend on the type of vehicle, the route chosen, and the number of shared bikes loaded during distribution. For example, if the demand stations distant to the distribution center are not promptly met, the public may switch to other transportation with lack of available shared bikes, which increases carbon emission. Therefore, the delivery center will allocate and deliver shared bikes according to the demand of different sites, and make decisions on the delivery route of the delivery vehicles and the distribution amount passing through each demand site, so as to reduce the carbon emission cost and delivery cost generated by the delivery vehicles.

In existing theoretical research, studies on shared bikes delivery vehicle route selection considering carbon emission under demand splitting have been scarce. Exploring the existing research results, one kind of research is demand splitting general cargo vehicle routing, which mainly make the minimum delivery cost as the goal without considering carbon emission, or take the minimum carbon emission as the target without considering delivery cost. The other kind is shared bikes delivery vehicle route selection, which pursues the minimum delivery cost, or the minimum number of vehicles and the maximum user satisfaction. However, the carbon emission generated by the delivery vehicles are not considered.

Aiming at the shortcomings of the existing research, the paper established the model, which takes the minimum sum of carbon emission cost and delivery cost as the objective to determine the route of the delivery vehicle and the volume of shared bikes to be delivered at each demand site, under demand splitting of the stations and considering carbon emission from the distribution vehicle. The algorithm was designed and its time complexity was analyzed. The effectiveness of the model and algorithm was verified through practical examples.

## 2. Literature review

The split delivery vehicle routing problem was proposed by Trudeau in 1989 [2]. When the demand is high, a single delivery of vehicles is not enough, forcing the supply to be split into multiple deliveries of vehicles [3] or a vehicle with multiple

deliveries [4]. The problem is then to determine the number of delivery vehicles or the number of vehicle departures, the delivery route of each vehicle, and the delivery quantity at each station given the objective of delivery cost minimization. The prior literature on the split delivery vehicle routing problem, in the case of the same delivery vehicles, is defined by variable approaches such as the design of tabu search algorithms to achieve the objective of minimizing delivery time based on a limited delivery center loading capacity [5], or the design of genetic algorithms [6], clustering algorithms, and simulated annealing algorithms [7] to minimize the vehicle distance under vehicle volume constraints, or the design of two-stage algorithms [8] or tabu search algorithms [9] to minimize delivery cost given station visit increases.

However, vehicles also produce carbon emission in delivery, so it is of great environmental value to extend the consideration of split delivery vehicle routing selection to carbon emission. There exist prior studies on the split delivery vehicle routing selection with consideration of carbon emission. For example, studies have built split delivery vehicle routing selection model with the objective of minimizing carbon emission cost, designed numerical experiments [10] or improved particle swarm algorithms [11], or designed two-stage heuristic algorithms based on tabu search to solve low-carbon split delivery vehicle routing selection optimization models with the objective of minimum carbon emission cost generated by transportation [12]. However, all these studies only consider carbon emission cost but do not include vehicle delivery cost. In addition, given multiple vehicle types, the low-carbon delivery vehicle routing optimization model given carbon emission cost minimization is established to determine the vehicle routing and the delivery quantity for each station by means of the tabu search algorithm to solve the model [13]. A mathematical model to minimize the carbon emission and delivery cost [14], an optimal distribution route model of fresh agricultural products considering the carbon emission cost with the goal of minimizing the total distribution cost [15] is constructed. considering the factors affecting carbon emissions such as vehicle travel distance, load and speed [16], the carbon trading mechanism [17], the MOGVRP model with vehicle use cost, cargo loss cost, fuel consumption and carbon emission cost, punishment cost and minimization, and customer satisfaction maximization as optimization objectives [18]. A coupled urban logistics collection and transportation system for two stages of urban waste transportation is designed [19] to waste-energy- economy problem, and a mixed integer programming model by considering the joint optimization of the routing-speed of crowd-shipping vehicles is developed [20]. Based on these, the optimization model with the lowest total cost as the objective function [21], the goal of minimizing the total cost of distribution [22], the minimization of the sum of the costs of vehicle use, fuel consumption and carbon emission and the maximization of the average satisfaction of customers as the optimization objectives [23] are proposed successively. All these studies consider carbon emission cost and vehicle delivery cost, but do not involve demand splitting.

In research on shared bikes multiple types split delivery vehicle routing selection, whether with an objective of minimizing delivery cost [24], minimizing the operator cost and user penalty simultaneously [25], or an objective of minimizing the number of vehicles for delivery and maximizing users' satisfaction [26], or an objective of comprehensive profit optimization introducing dynamic pricing strategy with negative prices [27], or a CA-based Stackelberg competition model in a competitive bike sharing market to seek the optimal bike allocation points [28], carbon emission is not considered.

In order to solve the problem of shared bikes vehicles routing selection with split delivery considering carbon emission, we develop a model with objectives of carbon emission cost minimization and delivery cost minimization, design an approximate algorithm $GA$ to solve the vehicle delivery route and the delivery quantity for each station, and analyze the effectiveness of the model and the algorithm. The time complexity of the approximate algorithm $GA$ is proven to be $O\left(n^2\right)$, and the upper and lower bounds of the approximate ratio of $GA$ are discussed. Finally, we also take the real shared bikes stations in the Yanta district of Xi'an in China as an empirical example to verify the effectiveness of the model and the algorithm. The results and con yield a theoretical basis for decision-makers to optimize the delivery of shared bikes.

Therefore, the model with the minimization of both carbon emission cost and delivery cost as the objective function in the study, and introducing parameter δ to quantify carbon emission reductions when shared bicycles are used instead of other transport modes under equivalent conditions, is different from previous models for shared bicycle distribution vehicle routing did not account for carbon emissions from delivery vehicles. Prior studies assumed that a single delivery could fully satisfy the demand of any station (i.e., each station being served exactly once would meet its complete demand). The revised model modifies this constraint such that serving each station at least once will satisfy the station's demand requirements.

## 3. Problem description and modeling

The un-directed network $G = (V, E)$, consisting of a set of stations $V = \{v_0, v_1, v_2, \cdots, v_n\}$ and a set of edges $E = \{e_{ij}\}$, is known. Which $v_0$ is the delivery center, $\{v_1, v_2, \cdots, v_n\}$ is $n$ stations of $G$, $c_{ij}$ is the vehicle delivery cost along the edge. There are $K$ vehicles in the delivery center. All vehicles start at $v_0$ to deliver shared bikes at station $D_i$, and then return to the delivery center. The problem is how to determine the vehicle delivery route and the delivery quantity at all passing stations given the objectives of minimizing carbon emission cost and delivery cost.

In developing the model, we make the following assumptions.

(1) The weight of the edge satisfies the triangle inequality;

(2) There are sufficient delivery vehicles.

We then establish the model as follows, combining the objective of the problem and all the constraints.

$$minF = \sum_{k=1}^{K} \left[ \left( \sum_{i=0}^{n} \sum_{j=1}^{n} x_{ij}^k \cdot d_{ij} \right) \cdot \omega \cdot \left( Q - \sum_{i=1}^{n} y_{i-1}^k \right) + \sum_{i=0}^{n} x_{i0}^k d_{i0} \rho^0 - \delta \cdot \sum_{i=1}^{n} y_i^k \right]$$

$$+ \sum_{k=1}^{K} \sum_{i=0}^{n} \sum_{j=0}^{n} c_{ij} x_{ij}^k \tag{1}$$

$$s.t. \sum_{j=1}^{n} x_{0j}^k = 1, (k = 1, 2, \cdots, K) \tag{2}$$

$$\sum_{i=1}^{n} x_{i0}^k = 1, (k = 1, 2, \cdots, K) \tag{3}$$

$$\sum_{k=1}^{K} \sum_{i=0}^{n} x_{ij}^k \geq 1, (j = 0, 1, 2, \cdots, n; i \neq j) \tag{4}$$

$$\sum_{i=1}^{n} x_{ij}^k = \sum_{i=1}^{n} x_{ji}^k, (k = 1, 2, \cdots, K; i \neq j) \tag{5}$$

$$y_i^k \leq D_i \sum_{j=1}^{n} x_{ij}^k, (i = 1, 2, \cdots, n; k = 1, 2, \cdots, K; i \neq j) \tag{6}$$

$$\sum_{i=0}^{n} y_i^k \leq Q, (k = 1, 2, \cdots, K) \tag{7}$$

$$\sum_{k=1}^{K} y_i^k = D_i, (i = 0, 1, 2, \cdots, n) \tag{8}$$

$$x_{ij}^k \in \{0, 1\}, (i = 0, 1, 2, \cdots, n; j = 0, 1, 2, \cdots, n; i \neq j; k = 1, 2, \cdots, K) \tag{9}$$

$$y_i^k \in N^*, (i = 0, 1, 2, \cdots n; k = 0, 1, 2, \cdots K) \tag{10}$$

$F$ denotes the carbon emission cost and delivery cost. $x_{ij}^k$ is a binary variable coded as 0 or 1. Specifically, if number $k$ vehicle runs from station $v_i$ to station $v_j$ along edge $e_{ij}$, then its value is 1, otherwise its value is 0. $y_i^k$ indicates the shared bikes delivery to station $v_i$ of number $k$ vehicle, and gives the distance $d_{ij}$ from $v_i$ to $v_j$. $k$ is the number of shared bikes of vehicle in the delivery, and $Q$ is the maximum loading capacity of the vehicle, and $D_i$ is the station demand. $\rho^*$ denotes the carbon emission cost per unit distance of vehicle with a full load, $\rho^0$ denotes the carbon emission cost per unit distance of vehicle with an empty load, and $\omega$ is the marginal carbon emission cost per additional shared-bike delivery by vehicles, $\delta$ is the carbon cost savings per unit distance from bike-sharing substitution.

Formula (1) is the objective function of carbon emission cost and delivery cost minimization, where part 1 is the carbon emission cost and part 2 is the delivery cost. Constraint (2) indicates that each vehicle must start from delivery center $v_0$, constraint (3) indicates that each vehicle must return to delivery center $v_0$, and constraint (4) indicates that each station can be served at least once or at most three times to satisfy demand. Constraint (5) denotes the conservation of flow, which means any vehicle which enters a station must also leave the station, and constraint (6) indicates that only when delivery vehicle $k$ passes through station $v_i$ which can be served by the vehicle. Constraint (7) indicates that the actual load of the vehicle shall not exceed its maximum loading capacity, constraint (8) indicates that the demands of all the stations are met.

## 4. Design of model solution algorithm

The shared bikes distribution vehicle routing selection with split delivery considering carbon emission constraints is an NP hard problem. We present an approximate algorithm for solving the routing selection problem.

### 4.1. Algorithm design

Under the scenario where stations demand can be split for delivery, when a station demand exceeds the maximum capacity of a delivery vehicle ($D_i \geq Q$), many distribution vehicles are required to serve these stations. Therefore, when designing the algorithm, the stations with a shared bikes demand that exceeds the vehicle's maximum loading capacity are given priority to be served partially, ensuring the remaining demand is less than the vehicle's maximum capacity. Subsequently, all stations are served to fully satisfy their demands. The goal is to determine the vehicle route and the delivery quantity at each station which are solved with the objectives of minimizing carbon emission cost and delivery cost.

Step 1, prioritize these stations with demand exceeding the vehicle's maximum loading capacity.

First, for the set of these stations whose demand is greater than or equal to the vehicle's maximum capacity ($D_i \geq Q$) is $V'$, and calculate $z$ of these stations in set $V'$. Then calculate the shortest distance from the distribution center to each

station $v_i'$ by Dijkstra's algorithm, and delivery vehicles to serve these stations is $I = \frac{\sum_{i=1}^{z} D_i}{Q}$. After completing delivery, the vehicles must return to the distribution center along the shortest route.

Step 2, delivery of shared bikes to all stations.

First of all, for the sets of all stations is $V$, find the shortest distance from the delivery center to each station of $V$, and sort them in ascending order of the shortest distance, then select the stations with the shortest route for service. If the total demand along the route is $\sum_{i=1}^{n} D_i \leq Q$, then select the station closest to this station for service from the set $V$, and recalculate the total demand along the route. If $\sum_{i=1}^{n} D_i \leq Q$, the next closest station in $V$ is added to the route, and this process repeats until all stations demand are satisfied. The delivery vehicle then returns to the distribution center along the shortest route from the last station. Once all station demands are satisfied, the delivery can be completed, and the actual delivery route of each vehicle and the actual delivery quantities to each station are output.

Based on the above approach, the algorithm *GA* is designed. The reason for using the algorithm GA is as follows.

(1) The algorithm *GA* maintains strict structural congruence with the problem's underlying assumptions [29], supporting split delivery without introducing virtual nodes or additional encoding, avoiding model bias and facilitates formal derivation (including feasibility analysis, complexity assessment, and cost decomposition).

(2) The algorithm *GA* exhibits strong determinism with minimal hyper-parameters, ensuring reproducible and auditable results, and reducing interference from randomness and parameter tuning on experimental internal validity, while enabling comprehensive sensitivity analysis.

(3) The algorithm *GA* focuses on interpretable decision rules built around shortest-distance and capacity-constrained computations, and with clearly defined theoretical complexity and concise implementation. It serves as the research supports to subsequent extensions while ensuring the transferability and verifiability of conclusions.

## 4.2 Algorithm GA

Step 1, Create an empty set as *V* and $V'$, for $Z = 0$. To any station $v_i$, if $D_i \geq Q$, then add $v_i$ to the set of $V'$, and calculate $z$ of the set $V'$, otherwise add $v_i$ to the set of $V$.

Step 2: If $V' = \varnothing$, then move to step 5; otherwise, calculate the shortest route $P_i$ from $v_0$ to each station of the set $V'$ by means of the Dijkstra algorithm, then add $P_i$ into the edge set of $P$, namely add the edge of $P_i$ to the corresponding set $E_i$, then add $E_i$ into the set of $E$, for $y_i^k = Q \left( k = 1, 2, \ldots, I, I = \frac{\sum_{i=1}^{n} D_i}{Q} \right)$ and $D_i = D_i - Q$.

Step 3: If $D_i < Q$, add all the stations $v_i$ of the set $V'$ to the set $V$, delete all the stations $v_i$ of the set $V'$, otherwise move to step 2.

Step 4: Find the shortest route $P_i$ from $v_0$ to each station $v_i$ of the set $V$ by means of the Dijkstra algorithm.

Step 5: Sort $P_i$ in ascending order of the shortest route, and write it down as a set $P_i'$. All the edges of $P_i'$ named $E_i'$, then add $P_i'$ into $P'$, and add $E_i'$ into $E'$.

Step 6: For $i = 1$.

Step 7: Calculate $\sum_{i=1}^{n} D_i$, all station demand of $P_i'$.

Step 8: For $k = I + 1$.

Step 9: If $\sum_{i=1}^{n} D_i < Q$, find a station closest to $v_i$ and record it as $v_{i+1}$, then add it to $E_i'$, record $\sum_{i=1}^{n} D_i = \sum_{i=1}^{n} D_{i+1}$ and return to step 9, otherwise move to next step.

Step 10: If $\sum_{i=1}^{n} D_i \geq Q$, for $y_i^k = D_i$, and select the vehicles $k$ to deliver shared bikes to the stations of $P_i'$. If $Q - \sum_{i=0}^{n} y_i^k = 0$, the vehicles return to the delivery center along the shortest route, and for $k = k + 1$. If $k < K$, return to step 9; otherwise return to step 11, otherwise move to the next step.

Step 11: Calculate the remaining demand of each station $D_i = D_i - y_i^k$.

Step 12: Delete $v_i$ of $V$. If $V = \varnothing$ turn to the next step, otherwise for $i = i + 1$ and return to step 8.

Step 13: Output the value of $x_{ij}^k$ and $y_i^k$.

Pseudocode of the approximation algorithm *GA* is shown as Table 1.

### 4.3 Time complexity of the algorithm GA

There are $n$ stations that demands shared bikes, and there are at most $n$ stations where its demands are $D_i \geq Q$, thus comparisons $o(n)$ at most are required to obtain $z$ (the quantity stations of $D_i \geq Q$) by step 1, and the Dijkstra algorithm is used to calculate $O\left(z^2\right)$ of the shortest routes $P_i'$ from $v_0$ to each station $v_i'$ of the set $V'$ by step 2; the step 2 and step 3 involve a loop execution no more than $O\left(z^2\right)$, and the Dijkstra algorithm is used to calculate $O\left(n^2\right)$ of the shortest routes $P_i$ from $v_0$ to each station $v_i$ of the set $V$ by step 4. Then the number of sorts is $n^2$ from step 5, and the number of calculations $\sum_{i=1}^n D_i$ is $O(n)$ from step 7, and the number of calculation cycles is $O\left(n^2\right)$ at most from step 9 to step 10, and the number of calculations $D_i = D_i - y_i^k$ is $O(n)$ at most from step 11, and the number of calculation cycles is $O\left(n^2\right)$ at most from step 7 to step 12. In conclusion, the paper gets theorem 1.

**Theorem 1** The complexity of the approximate algorithm for the shared bikes vehicles routing problem with split delivery under carbon emission constraints is $O\left(n^2\right)$, where $n$ denotes the quantity of customers stations.

### 4.4 Approximate ratio of GA

For $OPT(I)$ is the optimal value of carbon emission cost and delivery cost, $A(I)$ is the value of carbon emission cost and delivery cost achieved from the algorithm, then the approximate ratio of *GA* is $\frac{A(I)}{P(I)}$.

There are $k$ vehicles in the delivery center, and each vehicle delivers shared bikes to any station only once and returns to the delivery center after deliveries. So the vehicle delivery route is $k$ at most, and the delivery distance of each route is not less than $(n + 1)mind_{ij}$, then the delivery distance of each vehicle is not less than $(n + 1)mind_{ij}$, and the carbon emission cost of each vehicle returning to the delivery center is not less than $(n + 1)mind_{i0}\rho^0$. As $mind_{i0} = mind_{ij}$, the carbon emission cost of each vehicle returning to the delivery center is not less than $(n + 1)mind_{ij}\rho^0$, and the delivery cost of each vehicle is not less than $2(n + 1)minc_{ij}$. As $c_{ij} = \theta d_{ij}$, the delivery cost of each vehicle is not less than $2(n + 1)min\theta d_{ij}$, then get Lemma 1.

**Lemma 1** The lower bounds of $OPT(I)$, the optimal value of carbon emission cost and delivery cost are $OPT(I) \geq k \cdot \left[(n + 1)\left(\omega \cdot Q + \rho^0 + 2\theta\right)mind_{ij} - \delta \cdot Q\right]$, for the vehicles routing problem with split delivery considering carbon.

The proof is as follows.

The delivery distance of all vehicles is not less than $k \cdot (n + 1)mind_{ij}$, and the carbon emission cost of all vehicles returning to the delivery center is not less than $k \cdot (n + 1)mind_{ij}\rho^0$, and the delivery cost of all vehicles is not less than $2k \cdot \theta \cdot (n + 1)mind_{ij}$. Therefore, for the vehicle routing problem with split delivery considering carbon emission, the lower bounds of $OPT(I)$, the optimal value of carbon emission cost and delivery cost are $OPT(I) \geq k \cdot \left[(n + 1)\left(\omega \cdot Q + \rho^0 + 2\theta\right)mind_{ij} - \delta \cdot Q\right]$. Then Lemma 1 goes.

The algorithm *GA* is applied to any example, and the solution is as follows by *GA*.

$$A(I) = min[(\sum_{k=1}^K \sum_{i=0}^n \sum_{j=1}^n x_{ij}^k \cdot d_{ij}) \cdot w \cdot \sum_{k=1}^K (Q - \sum_{i=1}^n y_{i-1}^k)$$
$$-\delta \cdot \sum_{k=1}^K \sum_{i=1}^n y_i^k + \sum_{k=1}^K \sum_{i=0}^n x_{i0}^k d_{i0}\rho^0 + \sum_{k=1}^K \sum_{i=0}^n \sum_{j=0}^n c_{ij}x_{ij}^k]$$

$$\leq (I \cdot (n + 1)maxd_{ij} + k \cdot (n + 1 + k) \cdot maxd_{ij}) \cdot \omega \left(Q - \sum_{i=1}^n y_{i-1}\right) - k \cdot \delta \cdot \sum_{i=1}^n y_i$$

**Table 1. Pseudocode of the approximation algorithm *GA*.**

| Algorithm: *GA* |
|---|

Input:
- a set of stations V_all = {$v_0$, $v_1$, …, $v_n$}($v_0$ is the delivery center)
- demand of each station D[i](i = 1..n,D[0]=0)
- quantity of delivery vehicles K,the maximum loading capacity of the vehicle Q
- the shortest distance Matrix dist[·][·]

Output:
- $x_{ij}^k$(whether vehicle k traverses edge i→j)
- $y_i^k$(delivery quantity of vehicle k at station i)

// step1: set partitioning
V'←∅; V←∅                // V': stations of D[i] ≥ Q; V: stations of D[i] < Q
for i = 1..n do
if D[i] ≥ Q then V'←V' ∪ {v_i} else V←V ∪ {v_i} end if
end for

z←|V'|                // number of stations in V'
// step2–3: do full direct round trips for stations in V' Until D[i]<Q
while V'≠∅ do
 // calculate the shortest route P_i from v0 to each v_i∈V' by Dijkstra's algorithm, with edge set E_i
 for each v_i∈V' do
 P_i←Dijkstra($v_0$, v_i); E_i←edges(P_i)
 // do one full-load service
 choose a vehicle k (k ≤ K)
 $y_i^k$←Q; D[i] ← D[i] − Q
activate $x_{uv}^k$ = 1 for all edges (u,v) ∈ E_i ∪ reverse(E_i)// outbound trip and return trip
 end for
 // If D[i]<Q after this round of service, move the station to V
 for each v_i∈V' do
 if D[i] < Q then V←V ∪ {v_i} end if
 end for
 V' ← {v_i∈V_all \ {$v_0$} | D[i] ≥ Q}        // re-calculate
end while
// step 4–5: generate shortest routes to $v_0$ for stations in V and sort by ascending distance
P ← []; E' ← []
for each v_i∈V do
 P_i←Dijkstra($v_0$, v_i)
 append P←P_i; append E'←edges(P_i)
end for
P'←sort_by_length_ascending(P)        // sorted routes P'_1, P'_2, …
// step 6: extend routes sequentially to full a vehicle starting from i = 1
i←1
while i ≤ |P'| and (∃ D[j] > 0) do
// step 7: calculate the demands of all stations of P'_i
 S←sum_{v_j∈stations_on(P'_i)} D[j]
l←count_of_stations_on(P'_i)
 k←l+1
// step 8: initial route Pi'
// step 9: if not fully loaded, Iterativelymerge the mearest unserved stations from the last station v_l of P'_i
 while S < Q and (∃ u∈V with D[u] > 0 and u∉stations_on(P'_i)) do
 v_next ← argmin_{u} dist[v_l][u]
 merge v_next into the current route by connecting via its shortest edge
 P'_i←concatenate(P'_i, Dijkstra(v_l, v_next))
 E'_i←edges(P'_i)
 S←S + D[index(v_next)]
 l←l+1; k←k+1
 v_l←v_next
 end while
//step 10: distribute vehicle k along route P'_i when capacity exceeds Q
 remaining←Q

*(Continued)*

**Table 1.** (Continued)

```
    choose a vehicle k (k ≤ K)
for each station v_j in visit_order(P'_i) do
    y_j^k ← min(D[j], remaining)
    remaining ← remaining − y_j^k
if y_j^k > 0 then activate x along the segment edges to reach v_j end if
    if remaining = 0 then
        //activate x_{ij}^k = 1 for all edges on Dijkstra(v_j, v₀)
        break
        end if
    end for
// if remaining > 0 and more stations exist, continue extending route P_i',and return to step 9 and distribute
next vehicle (if needed)
// step 11: update remaining demands
for each v_j in V do D[j] ← D[j] − y_j^k end for
// step 12: remove fully served stations,and proceed to next shortest route
    V ← {v_j ∈ V | D[j] > 0}
    i ← i + 1
end while
// step 13: output
output x_{ij}^k, y_i^k
```

$$+k(n + 1)maxd_{i0}\rho^0 2 \left( I \cdot (n + 1)maxc_{ij} + k \cdot (n + 1 + k) \cdot maxc_{ij} \right)$$

$$= \left[ I \cdot (n + 1) + k^2 \right] maxd_{ij}\omega \cdot Q + 2 \left[ I \cdot (n + 1) + k^2 \right] \theta maxd_{ij}$$

$$+k \cdot \left[ (n + 1) \left( \omega \cdot Q + \rho^0 + 2\theta \right) maxd_{ij} - \delta \cdot Q \right]$$

$$= \left[ I \cdot (n + 1) + k^2 \right] (\omega \cdot Q + 2\theta)maxd_{ij} + k \cdot \left[ (n + 1) \left( \omega \cdot Q + \rho^0 + 2\theta \right) maxd_{ij} - \delta \cdot Q \right]$$

For $max\ d_{ij} = \gamma\ min\ d_{ij} = \gamma d$, according to the above analysis, then achieve the following theorem 2.

**Theorem 2** For the vehicle routing problem with split delivery considering carbon, the approximate ratio of *GA* is as follows.

$$\alpha = \gamma + \frac{\left[ I \cdot (n + 1) + k^2 \right] (\omega \cdot Q + 2\theta) \cdot \gamma + (\gamma - 1) \cdot \delta \cdot (Q/d)}{k \cdot \left[ (n + 1) \left( \omega \cdot Q + \rho^0 + 2\theta \right) - \delta \cdot (Q/d) \right]}$$

The above analysis shows that the approximate ratio $\alpha$ of *GA* is proportional to the number of *I* (the quantity of vehicles delivering shared bikes to the stations for which demand is larger than the maximum loading capacity *Q* of the vehicle), and inversely proportional to *Q*.

Combined with Theorem 2, the variation range of approximate ratio is further discussed and the following two inferences are given.

If $I = k(k > n)$, and $\forall v_i$, there exists $Q \leq D_i$, namely, the demand of each station is larger than the maximum loading capacity of the vehicle in the delivery center, then the value of the approximate ratio of *GA* reaches the maximum value.

$$\alpha = 1 + \frac{\left[l \cdot (n+1) + k^2\right](\omega \cdot Q + 2\theta) \cdot \gamma}{k \cdot \left[(n+1)(\omega \cdot Q + \rho^0 + 2\theta) - \delta \cdot \left(\frac{Q}{d}\right)\right]}$$

$$\leq \gamma + \frac{k[n+1+k](\omega \cdot min\ Q + 2\theta) \cdot \gamma + (\gamma-1)\delta \cdot \min(Q/d)}{k \cdot \left[(n+1)(\omega \cdot min\ Q + \rho^0 + 2\theta) - \delta \cdot \min(Q/d)\right]}$$

$$= \gamma + \frac{(n+1+k)(\omega \cdot min\ Q + 2\theta) \cdot \gamma - \delta \cdot min\ Q/d + \gamma \cdot \delta \cdot min(Q/d)}{(n+1)(\omega \cdot min\ Q + \rho^0 + 2\theta) - \delta \cdot min(Q/d)}$$

$$\leq \gamma + 1 + \frac{2k\theta \cdot \gamma}{(n+1) \cdot 2\theta}$$

$$\leq 1 + \left(1 + \frac{k}{n+1}\right) \cdot \gamma$$

Therefore, the approximate ratio of *GA* is $\alpha \leq 1 + (1 + \frac{k}{n+1}) \cdot \gamma, \gamma \in [\varphi, \eta](\varphi \leq \eta)$, then achieve corollary 1 as follows.

**Corollary 1** For the vehicle routing problem with split delivery considering carbon emission, the upper bound of the approximate ratio of *GA* is $1 + (1 + \frac{k}{n+1}) \cdot \eta$.

If $l = k = 1$, and $\forall v_i$, there exists $Q \leq D_i$, that is, the only one vehicle of the delivery center can satisfy the demand of all stations, the value $\alpha$ of the approximate ratio of *GA* reaches the minimum value.

$$\alpha = \gamma + \frac{\left[l \cdot (n+1) + k^2\right](\omega \cdot Q + 2\theta) \cdot \gamma + (\gamma-1)\delta \cdot max(\frac{Q}{d})}{k \cdot \left[(n+1)(\omega \cdot Q + \rho^0 + 2\theta) - \delta \cdot \max\left(\frac{Q}{d}\right)\right]}$$

$$\geq \gamma + \frac{(n+2)(\omega \cdot max\ Q + 2\theta) \cdot \gamma + (\gamma-1)\delta \cdot max(Q/d)}{(n+1)(\omega \cdot max\ Q + \rho^0 + 2\theta) - \delta \cdot \max(Q/d)})$$

$$\geq \gamma + \frac{(n+2)(\omega \cdot max\ Q + 2\theta) \cdot \gamma}{(n+1)(\omega \cdot max\ Q + \rho^0 + 2\theta)}$$

$$\geq \gamma + \frac{(n+2) \cdot \omega \cdot max\ Q}{(n+1) \cdot \omega \cdot max\ Q} \cdot \gamma$$

$$= \left(1 + \frac{n+2}{n+1}\right) \cdot \gamma$$

Therefore, the approximate ratio of *GA* is $\alpha \geq \left(1 + \frac{n+2}{n+1}\right) \cdot \gamma, \gamma \in [\varphi, \eta](\varphi \leq \eta)$, then achieve corollary 2 as follows.

**Corollary 2** For the multiple type vehicle routing problem with split delivery considering carbon emission, the lower bound of approximate ratio of *GA* is

$$\left(1 + \frac{n+2}{n+1}\right) \cdot \varphi$$

## 5. Case analysis

### 5.1 Operational case of shared bikes delivery in Yanta District of Xi'an

Yanta District is the largest core region in Xi'an, China, with rich cultural tourism resources such as the Wild Goose Pagoda, the Shaanxi Provincial Museum, the Grand Tang Dynasty Ever-bright City, Han Kiln, and South Lake. There are also core business zones such as Xiaozhai Saige and Dayuecheng. The region is so crowded that shared bikes usage increases year over year. The locations of 15 shared bike stations in Yanta District are shown in Fig 1, all the stations' daily demand for shared bikes is shown as Table 2. All stations location are derived from Amap's (Gaode Map) geospatial intelligence platform [30], and the demand data were provided by Annual report on urban traffic management in Xi'an [31].

There are 1,200 shared bikes in the delivery center, which are delivered by 20 vehicles with loading 60 shared bikes at most. In this study, the electric Foton Tuanao X5 is used as distribution vehicle [32]. The carbon emission cost with a full load per unit of distance $\rho^*$ is 0.0085RMB/km [33], and the carbon emission cost with an empty load per unit of distance $\rho^0$ is 0.0072 RMB/km [33]. If there is one more vehicle for delivery, the marginal carbon emission cost per additional shared-bike delivery by vehicles is $w$, and its value is 0.000021RMB/km [33]. The delivery cost per unit of distance $c_{ij}$ is 0.1RMB/km [34], the carbon cost savings per unit distance from bike-sharing substitution is $\delta$, and its value ranges between 0.3 and 0.5 [35, 36], and 0.5 is adopted in this study.

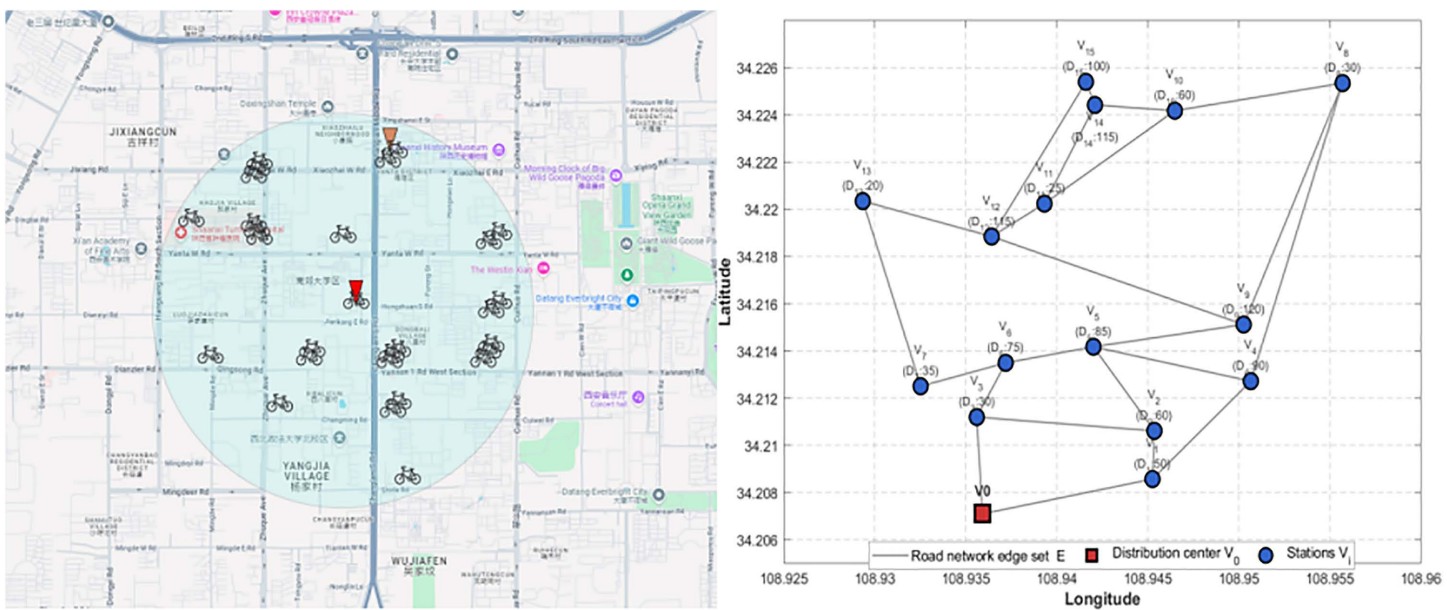

**Fig 1. Stations spatial distribution diagram.**

**Table 2. Stations location and demand.**

| ID | Name | Lng. | Lat. | Demand |
|---|---|---|---|---|
| 0 | Delivery center | 108.935914368 | 34.207106529 | |
| 1 | Xi'an International Studies University | 108.945249 | 34.20858033 | 50 |
| 2 | North Gate of Xi'an International Studies University | 108.945358526 | 34.210619397 | 60 |
| 3 | Vermilion Clouds Sky | 108.935600877 | 34.211208649 | 30 |
| 4 | Hanlin Shijia | 108.950656142 | 34.212728169 | 90 |
| 5 | Weiyi Street | 108.942007500 | 34.214182910 | 85 |
| 6 | Institute 213 | 108.937194362 | 34.213505089 | 75 |
| 7 | Rongshang District 10 | 108.932525769 | 34.212511229 | 35 |
| 8 | South Court of Yanta District | 108.955696668 | 34.225352881 | 30 |
| 9 | Changqing Fang | 108.950265256 | 34.215124212 | 120 |
| 10 | Super 8 Hotel | 108.9464819 | 34.22418725 | 60 |
| 11 | Provincial Cultural Relics Bureau | 108.939326828 | 34.220247861 | 25 |
| 12 | Medical School of Xi'an Jiaotong University | 108.936426031 | 34.218853552 | 115 |
| 13 | Haojia City Garden | 108.929340913 | 34.220355266 | 20 |
| 14 | Xiaozhai Subway Station | 108.942101375 | 34.224431088 | 115 |
| 15 | Customs Plaza | 108.941603800 | 34.225405700 | 100 |
| Total demand | | | | 1010 |

The daily shared bikes demand at each station is shown in Fig 2. The demand of shared bikes of 15 stations a day is 1010. The vehicles in the delivery center can satisfy demand, then the vehicles return to the delivery center after delivery. How to make decisions to vehicles routing and the quantity assigned to passing stations.

The proposed algorithm was applied to solve the case study, with the delivery process divided into two steps.

step1. Delivery to stations with demand exceeding vehicle capacity. The stations requiring more shared bikes than vehicle's maximum loading capacity ($Q=60$) include Stations 2, 4, 5, 6, 9, 10, 12, 14 and 15, then these stations were sorted

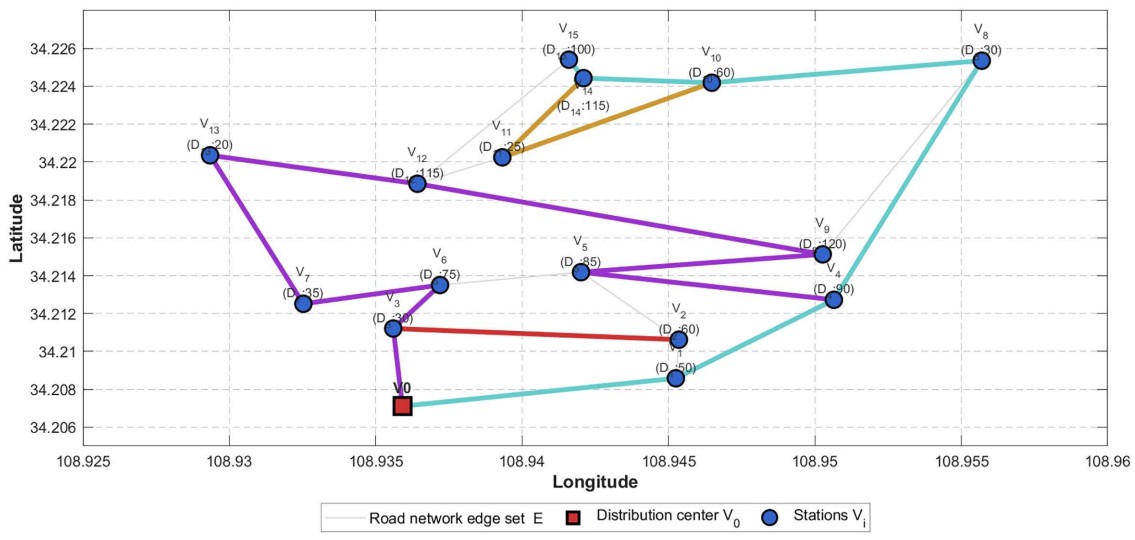

**Fig 2. shared bikes distribution vehicle routes diagram.**

in decreasing order of bikes demands as Stations 9, 12, 14, 15,4, 5, 6, 2 and10. The delivery vehicles allocated exactly 60 bikes to each of these stations.

Step2. Satisfy the demand of all stations. There are 600 shared bikes retained at the distribution center can satisfy a remaining station demand of 410.

Each vehicle route and the quantity of passing stations by the established model and the algorithm *GA* are shown in Table 3, the detailed vehicle routes are shown in Fig 2. The carbon emission and delivery cost are RMB 1515.76 in total, including carbon emission cost of RMB 504.12 and delivery cost of RMB 2019.88 after 1010 shared bikes are delivered by 17 vehicles. The approximate ratio is 3.52, which works well in practice.

## 5.2 Sensitivity analysis

**(1) Impact of *Q* on cost.** To isolate the effect of the vehicle maximum capacity (*Q*) on delivery cost and carbon emission cost, we conducted a ceteris paribus analysis by systematically varying *Q* while holding all other parameters constant. The sensitivity analysis examines how *Q* influences delivery cost and carbon emission cost. The maximum capacity of electric Foton Tuanao X5 is from 30 to 60, thus sensitivity analysis is performed with *Q* values of 30, 40, 50 and 60 in this study. As visualized in Fig 3, X-axis represents *Q* and Y-axis represents delivery cost and carbon emission cost.

From Fig 3, it can be seen that delivery cost and carbon emission cost exhibit a monotonically decreasing trend with increasing *Q*. This is because the increase in vehicle capacity enables the vehicle to transport more shared bikes each time, reducing the number of delivery vehicles and thus lowering the vehicle delivery cost and carbon emission cost.

**(2) Impact of *δ* on carbon emission cost.** To isolate the effect of the carbon cost savings/km from bike-sharing substitution (*δ*) on carbon emission cost, we conducted a ceteris paribus analysis by systematically varying *δ* while holding all other parameters constant. The sensitivity analysis examines how *δ* influences carbon emission cost. *δ* ranges

**Table 3. Vehicle routes and quantity assigned to passing stations.**

| Vehicles | Route | v1 | v2 | v3 | v4 | v5 | v6 | v7 | v8 | v9 | v10 | v11 | v12 | V13 | v14 | v15 |
|---|---|---|---|---|---|---|---|---|---|---|---|---|---|---|---|---|
| $k_1$ | $v_0 \to v_9$ | 0 | 0 | 0 | 0 | 0 | 0 | 0 | 0 | 60 | 0 | 0 | 0 | 0 | 0 | 0 |
| $k_2$ | $v_0 \to v_9$ | 0 | 0 | 0 | 0 | 0 | 0 | 0 | 0 | 60 | 0 | 0 | 0 | 0 | 0 | 0 |
| $k_3$ | $v0 \to v12$ | 0 | 0 | 0 | 0 | 0 | 0 | 0 | 0 | 0 | 0 | 0 | 60 | 0 | 0 | 0 |
| $k_4$ | $v0 \to v14$ | 0 | 0 | 0 | 0 | 0 | 0 | 0 | 0 | 0 | 0 | 0 | 0 | 0 | 60 | 0 |
| $k_5$ | $v0 \to v15$ | 0 | 0 | 0 | 0 | 0 | 0 | 0 | 0 | 0 | 0 | 0 | 0 | 0 | 0 | 60 |
| $k_6$ | $v0 \to v4$ | 0 | 0 | 0 | 60 | 0 | 0 | 0 | 0 | 0 | 0 | 0 | 0 | 0 | 0 | 0 |
| $k_7$ | $v0 \to v5$ | 0 | 0 | 0 | 0 | 60 | 0 | 0 | 0 | 0 | 0 | 0 | 0 | 0 | 0 | 0 |
| $k_8$ | $v0 \to v6$ | 0 | 0 | 0 | 0 | 0 | 60 | 0 | 0 | 0 | 0 | 0 | 0 | 0 | 0 | 0 |
| $k_9$ | $v0 \to v2$ | 0 | 60 | 0 | 0 | 0 | 0 | 0 | 0 | 0 | 0 | 0 | 0 | 0 | 0 | 0 |
| $k_{10}$ | $v0 \to v10$ | 0 | 0 | 0 | 0 | 0 | 0 | 0 | 0 | 0 | 60 | 0 | 0 | 0 | 0 | 0 |
| $k_{11}$ | $v0 \to v1 \to v4$ | 50 | 0 | 0 | 10 | 0 | 0 | 0 | 0 | 0 | 0 | 0 | 0 | 0 | 0 | 0 |
| $k_{12}$ | $v0 \to v3 \to v6 \to v7$ | 0 | 0 | 30 | 0 | 0 | 15 | 15 | 0 | 0 | 0 | 0 | 0 | 0 | 0 | 0 |
| $k_{13}$ | $v0 \to v4 \to v8 \to v11$ | 0 | 0 | 0 | 20 | 0 | 0 | 0 | 30 | 0 | 0 | 10 | 0 | 0 | 0 | 0 |
| $k_{14}$ | $v0 \to v7 \to v13 \to v12$ | 0 | 0 | 0 | 0 | 0 | 0 | 20 | 0 | 0 | 0 | 0 | 20 | 20 | 0 | 0 |
| $k_{15}$ | $v0 \to v15 \to v12$ | 0 | 0 | 0 | 0 | 25 | 0 | 0 | 0 | 0 | 0 | 0 | 35 | 0 | 0 | 0 |
| $k_{16}$ | $v0 \to v11 \to v14$ | 0 | 0 | 0 | 0 | 0 | 0 | 0 | 0 | 0 | 0 | 15 | 0 | 0 | 45 | 0 |
| $k_{17}$ | $v0 \to v14 \to v15$ | 0 | 0 | 0 | 0 | 0 | 0 | 0 | 0 | 0 | 0 | 0 | 0 | 0 | 10 | 40 |

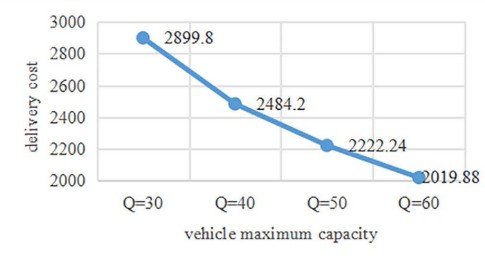 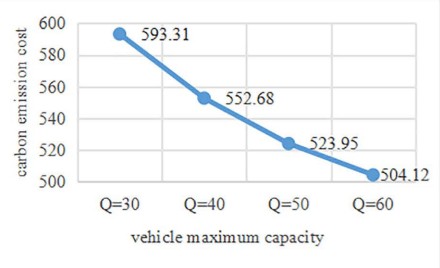

**Fig 3. Impact of Q on cost analysis diagram.**

between 0.3 and 0.5 [35,36], thus sensitivity analysis is performed with δ values of 0.3, 0.35, 0.4,0.45 and 0.5 in this study. As visualized in Fig 4, X-axis is δ and Y-axis is carbon emission cost.

From Fig 4, it can be seen that carbon emission cost exhibits a monotonically increasing trend with increasing δ. This is because the increase in the carbon cost savings per km from bike-sharing substitution enables the carbon emission cost increases even with consistent shared-bike delivery volumes at each time.

## 6. Conclusions

The problem of shared bikes distribution vehicles routing selection with split delivery considering carbon emission constraints is important. Exploring the existing research results, one kind of research is demand-split general cargo vehicle routing, which mainly take the minimization of delivery cost as the goal without considering carbon emission, or take the minimum carbon emission as the target without considering distribution costs; the other kind is shared bikes distribution vehicle route selection, which pursues the minimum delivery cost, or the minimum number of vehicles and the maximum user satisfaction. However, the carbon emission generated by the delivery vehicles is not considered. Aiming at the shortcomings of the existing research, we establish a vehicle routing selection model of split delivery carbon emission with the objectives of minimizing carbon emission cost and delivery cost, designs the approximation algorithm *GA and* proves the time complexity of $GA\ O\left(n^2\right)$ (Where $x_2$ denotes demand stations) of *GA,* then discusses the upper and lower bounds of approximate ratio. Finally, we take real shared bikes stations in the Yanta district of Xi'an in China as an empirical example to verify the effectiveness of the model and the algorithm. The results can help to provide a theoretical basis for decision-makers to optimize the delivery of shared bikes.

The research has made some gains, but it can do further research on the existing basis, such as the assumption that the number of distribution vehicles cannot meet the demand of distribution or the demand of shared bikes is variable.

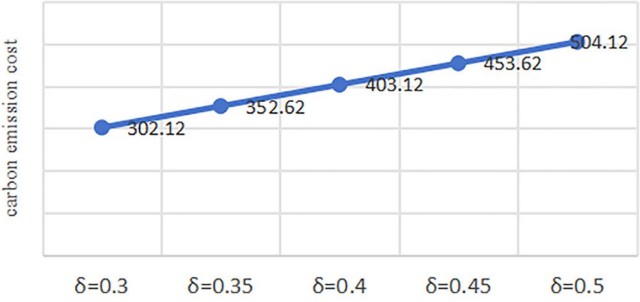

**Fig 4. Impact of δ on carbon emission cost analysis diagram.**

## Author contributions

**Formal analysis:** Qinge Guo.

**Investigation:** Qinge Guo.

**Methodology:** Qinge Guo, Hao Ji.

**Writing – original draft:** Guanghui Chen, Huicong Li, Bing Su.

**Writing – review & editing:** Hao Ji.

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
