## [Decision Letter · Decision Letter 0]

7 Jul 2025

PONE-D-25-24555Shared Bike Distribution Vehicle Routing Problem with Split Delivery Considering Carbon EmissionPLOS ONE

Dear Dr. Chen,

Thank you for submitting your manuscript to PLOS ONE. After careful consideration, we feel that it has merit but does not fully meet PLOS ONE’s publication criteria as it currently stands. Therefore, we invite you to submit a revised version of the manuscript that addresses the points raised during the review process.

Dear Author,

The reviewers have recommended major revisions for your manuscript. Please address the comments and suggestions provided by the reviewers thoroughly in your revision.

We believe that these changes will significantly improve the quality and impact of your work. We look forward to receiving your revised manuscript.

Best regards

We look forward to receiving your revised manuscript.

Kind regards,

Manuel Herrador, Ph.D.

Academic Editor

PLOS ONE

 [This work was supported in part by the National Nature Science Foundation of China under Grant 52302443, and in part by the Social Science Foundation of Shaanxi Province under Grant 2023R059.]. 

Additional Editor Comments:

Dear Author,

The reviewers have recommended major revisions for your manuscript. Please address the comments and suggestions provided by the reviewers thoroughly in your revision.

We believe that these changes will significantly improve the quality and impact of your work. We look forward to receiving your revised manuscript.

Best regards

Reviewers' comments:

Reviewer's Responses to Questions

**Comments to the Author**

1. Is the manuscript technically sound, and do the data support the conclusions?

Reviewer #1: Partly

Reviewer #2: Yes

2. Has the statistical analysis been performed appropriately and rigorously? 

Reviewer #1: No

Reviewer #2: Yes

3. Have the authors made all data underlying the findings in their manuscript fully available?

Reviewer #1: Yes

Reviewer #2: Yes

4. Is the manuscript presented in an intelligible fashion and written in standard English?

Reviewer #1: Yes

Reviewer #2: Yes

5. Review Comments to the Author

Reviewer #1: The manuscript addresses an important and timely topic: the optimization of shared bike distribution vehicle routing under split delivery constraints while considering carbon emissions. The problem is relevant for sustainable urban logistics and aligns with current efforts to reduce the environmental impact of transportation. The proposed model and approximation algorithm (GA) are well-structured, and the empirical validation provides a strong application to real-world scenarios. However, there are several critical areas where the manuscript requires improvement before it can be considered for publication.

1. While the manuscript provides an extensive list of references, the literature review lacks critical synthesis. The discussion on the gaps in existing research is vague and does not clearly articulate how this work advances the field. Please compare the proposed model and algorithm to existing models/algorithms for split delivery vehicle routing.

2. Please discuss how the carbon emission considerations in this study differ from prior studies.

3. The description of the approximation algorithm (GA) is difficult to follow. The steps of the algorithm are not sufficiently detailed for replication. Please provide a pseudocode or flowchart for the GA algorithm.

4. Please justify why this specific algorithm was chosen over other heuristic or metaheuristic approaches (e.g., tabu search, particle swarm optimization).

5. Please elaborate on the parameter settings for the GA and their impact on the results.

6. Please discuss the source of the data (e.g., bike demand, vehicle characteristics) and any assumptions made during data collection and analysis.

7. Please include a visual representation of the vehicle routes (e.g., maps or diagrams) to make the results more intuitive.

8. The calculation of carbon emission costs is not sufficiently detailed. It is unclear how these costs are derived and whether they are based on real-world data or assumptions. Please provide more information about the methodology used to estimate carbon emissions per unit of distance/load.

9. The discussion of limitations is superficial. The authors should address the assumption of sufficient delivery vehicles may not hold in real-world scenarios. Discuss how the model would perform under vehicle shortages.

Reviewer #2: The paper aims to establish model of multiple types distribution vehicle route selection in shared bike of considering carbon emission. I have some major comments.

1. This paper is indeed working on the topic of relocation/repositioning of shared bikes. There is a large body of similar works in this topic. What is the major contribution of this study. Simply consider the carbon emission?

2. This work simply focuses on the operator-based relocation of shared bikes. However, user based relocation is also proposed in the literature to incentivize users to help do the relocation, which do not involve any carbon emission. One paper that you can refer to is entitled with "A dynamic pricing scheme with negative prices in dockless bike sharing systems" published on Transportation Research Part B.

3. The literature review of this study is NOT comprehensive. Many related works on the topic of bike sharing system are not mentioned and well reviewed. Some papers with the following titles are to be included in the literature review: "A data-driven dynamic repositioning model in bicycle-sharing systems", "Allocation strategies in a dockless bike sharing system: a community structure-based approach", "Optimal bike allocations in a competitive bike sharing market"; "Research on rebalancing of large-scale bike-sharing system driven by zonal heterogeneity and demand uncertainty".

4. The English language of this paper manuscript needs to be further polished and refined. There are many errors and language problems in this manuscript.

6. PLOS authors have the option to publish the peer review history of their article (what does this mean? ). If published, this will include your full peer review and any attached files.

**Do you want your identity to be public for this peer review?** For information about this choice, including consent withdrawal, please see our Privacy Policy .

Reviewer #1: No

Reviewer #2: No

---

## [Author Response · Author response to Decision Letter 1]

8 Sep 2025

2. Please note that PLOS ONE has specific guidelines on code sharing for submissions in which author-generated code underpins the findings in the manuscript. In these cases, we expect all author-generated code to be made available without restrictions upon publication of the work. Please review our guidelines at https://journals.plos.org/plosone/s/materials-and- software- sharing#loc-sharing-code and ensure that your code is shared in a way that follows best practice and facilitates reproducibility and reuse.

[This work was supported in part by the National Nature Science Foundation of China under Grant 52302443, and in part by the Social Science Foundation of Shaanxi Province under Grant 2023R059.].

Author response: Thank you for pointing this out. We have revised this statement. The revised content is as follows: This work was supported in part by the Social Science Foundation of Shaanxi Province under Grant S2023R059; 2025JC-YBMS-809.The funder had no role in study design, data collection and analysis, decision to publish, and preparation of the manuscript.

Review Comments to the Author

Reviewer #1: The manuscript addresses an important and timely topic: the optimization of shared bike distribution vehicle routing under split delivery constraints while considering carbon emissions. The problem is relevant for sustainable urban logistics and aligns with current efforts to reduce the environmental impact of transportation. The proposed model and approximation algorithm (GA) are well-structured, and the empirical validation provides a strong application to real-world scenarios. However, there are several critical areas where the manuscript requires improvement before it can be considered for publication.

Author response: Thank you!

1.While the manuscript provides an extensive list of references, the literature review lacks critical synthesis. The discussion on the gaps in existing research is vague and does not clearly articulate how this work advances the field. Please compare the proposed model and algorithm to existing models/algorithms for split delivery vehicle routing.

Author response: Thank you for pointing this out. We have revised the literature review by removing some identical studies on split vehicle routing problems and adding references related to bike-sharing delivery, with clarifications the differences between this work and existing literature. Please see line 96-103 in the revised manuscript.

Please discuss how the carbon emission considerations in this study differ from prior studies.

Author response: Thank you for pointing this out. We think this study focuses on the operator-based distribution vehicle routing of shared bikes, considering the significant demand variations across different stations, transforms the distribution vehicle routing of shared bikes problem into a Split Delivery Vehicle Routing Problem (SDVRP). However, a literature review reveals that most existing research on SDVRP focuses on minimizing costs, maximizing profits, or reducing the number of vehicles, while only a few studies incorporate carbon emissions into the objective function. Therefore, this paper takes this gap as a key research and conducts an in-depth investigation, and established the model with the minimization of both carbon emission cost and delivery cost as the objective functionin in the study, while introducing parameter δ to quantify carbon emission reductions when shared bicycles are used instead of other transport modes under equivalent conditions. We have added the content to the manuscript on line 113-119 in the revised manuscript.

3.The description of the approximation algorithm (GA) is difficult to follow. The steps of the algorithm are not sufficiently detailed for replication. Please provide a pseudocode or flowchart for the GA algorithm.

Author response: We think this is an excellent suggestion. We have added the suggested pseudocode to the manuscript on TABLE 1. Pseudocode of the approximation algorithm GA.

4.Please justify why this specific algorithm was chosen over other heuristic or metaheuristic approaches (e.g., tabu search, particle swarm optimization).

Author response: We have added the suggested content to the manuscript on line 185-197 in the revised manuscript.

5.Please elaborate on the parameter settings for the GA and their impact on the results.

Author response: Thank you for pointing this out. We have added the suggested content to the manuscript on 5.2 Sensitivity Analysis.

6. Please discuss the source of the data (e.g., bike demand, vehicle characteristics) and any assumptions made during data collection and analysis.

Author response: We have added the suggested content to the manuscript on line 312-336 and REFERENCES 30-36 in the revised manuscript.

7.Please include a visual representation of the vehicle routes (e.g., maps or diagrams) to make the results more intuitive.

Author response: We have added the suggested content as FIGURE 2 in the revised manuscript.

8.The calculation of carbon emission costs is not sufficiently detailed. It is unclear how these costs are derived and whether they are based on or assumptions. Please provide more information about the methodology used to estimate carbon emissions per unit of distance/load.

Author response: Thank you for pointing this out. We have added the suggested content to the manuscript on line 312-336 and REFERENCES 30-36 in the revised manuscript. The calculation of carbon emission cost is primarily based on real-world data, and follows a three-tiered methodology, First, the emission factor of electric of Foton Tuanao X5 is derived from empirical data, which then yields the per-unit-distance/per-unit-load carbon emission rate. Subsequently, carbon cost are obtained by applying jurisdictional carbon pricing schemes.

9.The discussion of limitations is superficial. The authors should address the assumption of sufficient delivery vehicles may not hold in real-world scenarios. Discuss how the model would perform under vehicle shortages.

Author response: We agree that this is a potential limitation of the study. We have added this as a limitation on line 428 in the revised manuscript. Our next study will specifically address the distribution vehicle routing of shared bikes under limited vehicle availability.

Reviewer #2: The paper aims to establish model of multiple types distribution vehicle route selection in shared bike of considering carbon emission. I have some major comments.

1. This paper is indeed working on the topic of relocation/repositioning of shared bikes. There is a large body of similar works in this topic. What is the major contribution of this study. Simply consider the carbon emission?

Author response: We appreciate the reviewer's feedback, we respectfully disagree. We think this paper focuses on the operator-based distribution vehicle routing of shared bikes, considering the significant demand variations across different stations, transforms the distribution vehicle routing of shared bikes problem into a Split Delivery Vehicle Routing Problem (SDVRP). However, a literature review reveals that most existing research on SDVRP focuses on minimizing costs, maximizing profits, or reducing the number of vehicles, while only a few studies incorporate carbon emissions into the objective function. Therefore, this paper took this gap as a key research and conducts an in-depth investigation, and established the model with the minimization of both carbon emission cost and delivery cost as the objective function in the study, while introducing parameter δ to quantify carbon emission reductions when shared bicycles are used instead of other transport modes under equivalent conditions. We have added the content to the manuscript on line 113-119 in the revised manuscript.

2. This work simply focuses on the operator-based relocation of shared bikes. However, user based relocation is also proposed in the literature to incentivize users to help do the relocation, which do not involve any carbon emission. One paper that you can refer to is entitled with "A dynamic pricing scheme with negative prices in dockless bike sharing systems" published on Transportation Research Part B.

Author response: Thank you for this suggestion. After thoroughly studying the literature cited by the reviewer, we found it very inspiring and believes it would have been interesting to explore this aspect. However, in our study, this would not be possible because that our work only focused on distribution vehicle routing of shared bikes and users participation in the redistribution of shared bikes was not taken into consideration.

3. The literature review of this study is NOT comprehensive. Many related works on the topic of bike sharing system are not mentioned and well reviewed. Some papers with the following titles are to be included in the literature review: "A data-driven dynamic repositioning model in bicycle-sharing systems", "Allocation strategies in a dockless bike sharing system: a community structure-based approach", "Optimal bike allocations in a competitive bike sharing market"; "Research on rebalancing of large-scale bike-sharing system driven by zonal heterogeneity and demand uncertainty".

Author response: Thank you for pointing this out. We have studied the literature recommended by the reviewers and supplemented the literature review accordingly, Please see line 96-103 in the revised manuscript.

4. The English language of this paper manuscript needs to be further polished and refined. There are many errors and language problems in this manuscript.

Author response: We agree with the reviewer's assessment. we have revised spelling and grammatical mistake in the revised manuscript.

---

## [Decision Letter · Decision Letter 1]

18 Sep 2025

Shared Bike Distribution Vehicle Routing Problem with Split Delivery Considering Carbon Emission

PONE-D-25-24555R1

Dear Dr. Chen,

We’re pleased to inform you that your manuscript has been judged scientifically suitable for publication and will be formally accepted for publication once it meets all outstanding technical requirements.

Kind regards,

Manuel Herrador, Ph.D.

Academic Editor

PLOS ONE

Additional Editor Comments (optional):

Dear authors,

I am pleased to inform you that your manuscript has been accepted for publication in PLOS ONE. Both reviewers recommended acceptance, and we appreciate the quality and significance of your work.

Thank you for choosing PLOS ONE as the platform to share your research. We look forward to working with you through the final stages of production.

Best regards

Reviewers' comments:

Reviewer's Responses to Questions

**Comments to the Author**

1. If the authors have adequately addressed your comments raised in a previous round of review and you feel that this manuscript is now acceptable for publication, you may indicate that here to bypass the “Comments to the Author” section, enter your conflict of interest statement in the “Confidential to Editor” section, and submit your "Accept" recommendation.

Reviewer #1: All comments have been addressed

Reviewer #2: All comments have been addressed

2. Is the manuscript technically sound, and do the data support the conclusions?

Reviewer #1: Yes

Reviewer #2: Yes

3. Has the statistical analysis been performed appropriately and rigorously? 

Reviewer #1: Yes

Reviewer #2: Yes

4. Have the authors made all data underlying the findings in their manuscript fully available?

Reviewer #1: Yes

Reviewer #2: Yes

5. Is the manuscript presented in an intelligible fashion and written in standard English?

Reviewer #1: Yes

Reviewer #2: Yes

6. Review Comments to the Author

Reviewer #1: Thank you for your revision. No further suggestions from me. The current version can be acceptable.

Reviewer #2: My comments have been addressed. I recommend to accept this paper manuscript for publication consideration in this journal.

7. PLOS authors have the option to publish the peer review history of their article (what does this mean? ). If published, this will include your full peer review and any attached files.

**Do you want your identity to be public for this peer review?** For information about this choice, including consent withdrawal, please see our Privacy Policy .

Reviewer #1: No

Reviewer #2: No

---

## [Editor Report · Acceptance letter]

PONE-D-25-24555R1

PLOS ONE

Dear Dr. Chen,

I'm pleased to inform you that your manuscript has been deemed suitable for publication in PLOS ONE. Congratulations! Your manuscript is now being handed over to our production team.

Kind regards,

on behalf of

Dr. Manuel Herrador

Academic Editor

PLOS ONE